# CONTRASTIVE LEARNING IS SPECTRAL CLUSTERING ON SIMILARITY GRAPH

**Zhiquan Tan**[1*]    **Yifan Zhang**[2*]    **Jingqin Yang**[2*]     **Yang Yuan**[2,3,4†]
[1]Department of Mathematical Sciences, Tsinghua University [2]IIIS, Tsinghua University
[3]Shanghai Artificial Intelligence Laboratory [4]Shanghai Qizhi Institute
{tanzq21, zhangyif21, yangjq21}@mails.tsinghua.edu.cn,
yuanyang@tsinghua.edu.cn

## ABSTRACT

Contrastive learning is a powerful self-supervised learning method, but we have a limited theoretical understanding of how it works and why it works. In this paper, we prove that contrastive learning with the standard InfoNCE loss is equivalent to spectral clustering on the similarity graph. Using this equivalence as the building block, we extend our analysis to the CLIP model and rigorously characterize how similar multi-modal objects are embedded together. Motivated by our theoretical insights, we introduce the Kernel-InfoNCE loss, incorporating mixtures of kernel functions that outperform the standard Gaussian kernel on several vision datasets[‡].

## 1 INTRODUCTION

Contrastive learning has emerged as one of the most prominent self-supervised learning methods, especially in the realm of vision tasks (Chen et al., 2020a; He et al., 2019b). This approach trains a neural network to map a set of objects into an embedding space, ensuring that similar objects are closely positioned while dissimilar objects remain distanced. The InfoNCE loss, exemplified by SimCLR (Chen et al., 2020a), is a widely employed loss function for achieving this goal.

In their inspiring work, HaoChen et al. (2021) demonstrated that by replacing the standard InfoNCE loss with their spectral contrastive loss, contrastive learning performs spectral clustering on the population augmentation graph. However, the spectral contrastive loss is seldom utilized in practice and is not applicable for analyzing the performance of various similarity functions in the embedding space. Furthermore, when employing the spectral contrastive loss, the final embedding constitutes a combination of standard spectral clustering and an additional linear transformation. Consequently, existing results do not establish a connection between the original InfoNCE loss and standard spectral clustering.

In this paper, we prove that SimCLR, the standard contrastive learning method, performs spectral clustering without modifying the InfoNCE loss or applying additional transformations to the embeddings. Our analysis involves a collection of $n$ objects $\mathbf{X} = [\mathbf{X}_1, \cdots, \mathbf{X}_n]$ within space $\mathcal{X}$. For these objects, we define a similarity graph with an adjacency matrix $\boldsymbol{\pi}$, such that $\boldsymbol{\pi}_{i,j}$ represents the probability of $\mathbf{X}_i$ and $\mathbf{X}_j$ being paired together in the data augmentation step of contrastive learning. Notice that $\boldsymbol{\pi}$ can be in general asymmetric.

Given this similarity graph, we aim to discover an embedding function $f : \mathcal{X} \to \mathcal{Z}$. Denote $\mathbf{Z} \triangleq f(\mathbf{X})$ as the embedding of $\mathbf{X}$, and our objective is to ensure that the Gram matrix $\mathbf{K_Z}$ with kernel $k$ representing the similarities for $\mathbf{Z}$ closely approximates $\boldsymbol{\pi}$. Please refer to Figure 1 for an illustration.

However, directly comparing $\boldsymbol{\pi}$ with $\mathbf{K_Z}$ can be difficult, as there are too many edges in both graphs. Therefore, we define two Markov random fields (MRFs) based on $\boldsymbol{\pi}$ and $\mathbf{K_Z}$ and compare the MRFs instead. Each MRF introduces a probability distribution of unweighted directed subgraphs on $n$

---

*Equal contribution.
†Corresponding author.
‡The code is available at https://github.com/yifanzhang-pro/Kernel-InfoNCE.

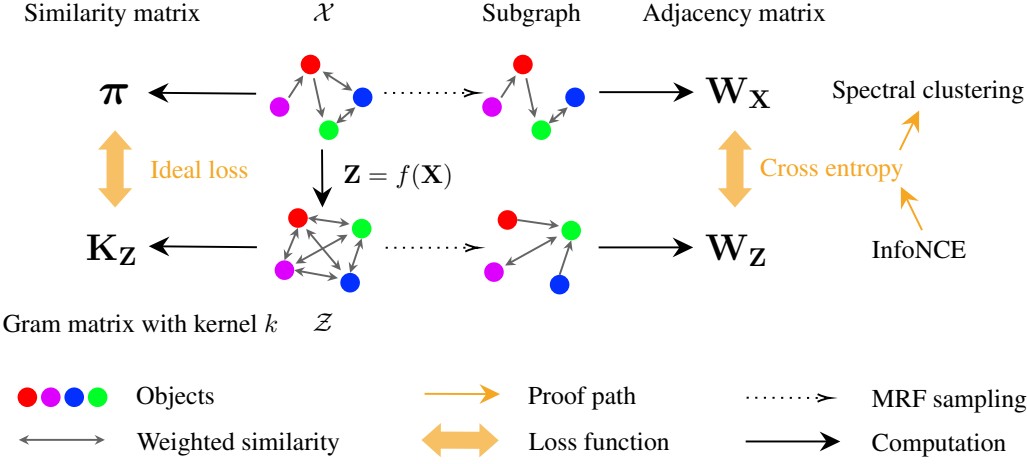

Figure 1: An illustration of our analysis. The similarity matrix $\boldsymbol{\pi}$ encapsulates the relationships between various images. Given the large size of the matrix, we employ a technique known as Markov Random Field sampling to sidestep the issue of direct utilization. Through our research, we discovered an equivalence between InfoNCE loss and a method known as spectral clustering when a Gaussian kernel function was utilized, thus validating our approach.

objects (Van Assel et al., 2022), denoted as $\mathbf{W_X}$ and $\mathbf{W_Z}$ respectively. As a natural approximation to the ideal loss between $\boldsymbol{\pi}$ and $\mathbf{K_Z}$, we employ the cross-entropy loss between $\mathbf{W_X}$ and $\mathbf{W_Z}$. Our paper's surprising discovery is that the InfoNCE loss is equivalent to the cross-entropy loss when each subgraph is constrained to have an out-degree of exactly one. Furthermore, when $k$ is the Gaussian kernel, optimizing the cross-entropy loss corresponds to executing spectral clustering on $\boldsymbol{\pi}$. By combining these two observations, we conclude that employing the InfoNCE loss is equivalent to performing spectral clustering.

Our characterization of contrastive learning hinges on two crucial factors: the augmentation step that defines a similarity graph, and the InfoNCE loss that measures the distance between two MRFs. Consequently, any other models incorporating these two factors can be similarly analyzed. Notably, the CLIP (Radford et al., 2021) model for multi-modal learning fits within this paradigm. Utilizing the same framework, we establish a representation theorem for CLIP, demonstrating that it performs spectral clustering on the bipartite graph induced by the paired training data.

Is it possible to improve the InfoNCE loss by using a different kernel? Based on the maximum entropy principle, we demonstrate that the exponential kernels are the natural choices for capturing the local similarity structure for contrastive learning. Empirically, we observe that taking the mixture of Gaussian and Laplacian kernels, which maintain the aforementioned properties, can achieve better performance than the Gaussian kernel on several benchmark vision datasets.

In summary, our main contributions include:

- We prove the equivalence of SimCLR and spectral clustering on the similarity graph.
- We extend our analysis to the multi-modal setting and prove the equivalence of CLIP and spectral clustering on the multi-modal similarity graph.
- Inspired by theory, we propose a new Kernel-InfoNCE loss with mixture of kernel functions that achieves better performance than the standard Gaussian kernel (SimCLR) empirically on the benchmark vision datasets.

## 2 BACKGROUND

In this section, we will introduce the basic knowledge we will use throughout the paper. In this paper, we use objects to denote data points like images or texts. Given a matrix $\mathbf{X}$, we use $\mathbf{X}_i$ to denote its $i$-th row, and $\mathbf{X}_{i,j}$ to denote its $(i,j)$-th entry. Same holds for matrices like $\mathbf{W_X}$, where we use $\mathbf{W}_{\mathbf{X},i}$ and $\mathbf{W}_{\mathbf{X},i,j}$, respectively.

## 2.1 Contrastive learning: SimCLR

Given a query object $\mathbf{q} \in \mathcal{X}$, **one** similar object (positive samples) $\mathbf{p}_1$ for $\mathbf{q}$, and $N-1$ other objects $\{\mathbf{p}_i\}_{i=2}^N$, SimCLR finds a function $\boldsymbol{f}$ (usually a neural network) that maps these objects to $\mathcal{Z}$, to minimize the InfoNCE loss of $\mathbf{q}$:

$$\mathcal{L}(\mathbf{q}, \mathbf{p}_1, \{\mathbf{p}_i\}_{i=2}^N) = -\log \frac{\exp(\mathrm{sim}(\boldsymbol{f}(\mathbf{q}), \boldsymbol{f}(\mathbf{p}_1))/\tau)}{\sum_{i=1}^N \exp(\mathrm{sim}(\boldsymbol{f}(\mathbf{q}), \boldsymbol{f}(\mathbf{p}_i))/\tau)} \tag{1}$$

Here, the actual loss of $\boldsymbol{f}$ takes the summation over different $\mathbf{q}$s, and $\tau$ is a temperature hyperparameter. The $\mathrm{sim}(\mathbf{Z}_i, \mathbf{Z}_j)$ function measures the similarity between $\mathbf{Z}_i, \mathbf{Z}_j$ in $\mathcal{Z}$, and is commonly defined as $\mathrm{sim}(\mathbf{Z}_i, \mathbf{Z}_j) = \frac{\mathbf{Z}_i^\top \mathbf{Z}_j}{\|\mathbf{Z}_i\|\|\mathbf{Z}_j\|}$, or $\mathbf{Z}_i^\top \mathbf{Z}_j$, or $-\|\mathbf{Z}_i - \mathbf{Z}_j\|^2/2$. In this paper, we consider the case that $\mathcal{Z}$ is the unit sphere, i.e., $\|\mathbf{Z}_i\| = \|\mathbf{Z}_j\| = 1$. This is because both SimCLR and CLIP have a normalization step in the implementation (Chen et al., 2020a; Radford et al., 2021). Hence, $\frac{\mathbf{Z}_i^\top \mathbf{Z}_j}{\|\mathbf{Z}_i\|\|\mathbf{Z}_j\|} = \mathbf{Z}_i^\top \mathbf{Z}_j$, and

$$-\|\mathbf{Z}_i - \mathbf{Z}_j\|^2/2 = -\mathbf{Z}_i^2/2 - \mathbf{Z}_j^2/2 + \mathbf{Z}_i^\top \mathbf{Z}_j = -1 + \mathbf{Z}_i^\top \mathbf{Z}_j. \tag{2}$$

Therefore, these losses are the same up to a constant.

## 2.2 Multi-modal learning: CLIP

CLIP (Radford et al., 2021) is a multi-modal model with a dataset containing millions of (image, text) pairs. During pretraining, for each batch of $N$ pairs of data points, CLIP uses an image encoder and a text encoder to get $N$ pairs of embeddings, and uses the InfoNCE loss to compute the correct $N$ pairs out of $N \times N$ possible connections. Specifically, given an image $\mathbf{a}_i$, we compare the its matching score of the paired text $\mathbf{b}_i$, with the matching scores of other $N-1$ texts $\{\mathbf{b}_j\}_{j \neq i}$, using the loss $\mathcal{L}(\mathbf{a}_i, \mathbf{b}_i, \{\mathbf{b}_j\}_{j \neq i})$ defined in Eqn. (1) by setting $\mathrm{sim}(\mathbf{Z}_i, \mathbf{Z}_j) = \frac{\mathbf{Z}_i^\top \mathbf{Z}_j}{\|\mathbf{Z}_i\|\|\mathbf{Z}_j\|}$. One can define the loss similarly for text, and the actual loss of the embedding network $\boldsymbol{f}$ takes the summation over all the images and texts.

## 2.3 Reproducing Kernel Hilbert Space

Given two objects $\mathbf{Z}_i, \mathbf{Z}_j \in \mathcal{Z}$, consider a feature map $\varphi : \mathcal{Z} \to \mathcal{H}$, where the feature space $\mathcal{H}$ is usually much larger than $\mathcal{Z}$. We may define a kernel $k$ that measures the similarity of $\mathbf{Z}_i, \mathbf{Z}_j$ as $k(\mathbf{Z}_i, \mathbf{Z}_j) \triangleq \langle \varphi(\mathbf{Z}_i), \varphi(\mathbf{Z}_j) \rangle_\mathcal{H}$, i.e., the inner product between the two objects after mapping them to the feature space. For any vector $h \in \mathcal{H}$, it also corresponds to a function $h(\cdot) : \mathcal{Z} \to \mathbb{R}$, defined as $h(\mathbf{Z}_i) = \langle h, \varphi(\mathbf{Z}_i) \rangle_\mathcal{H}$. Specifically, $\varphi(\mathbf{Z}_j)$ as a vector in $\mathcal{H}$ represents the function $k(\cdot, \mathbf{Z}_j) : \mathcal{Z} \to \mathbb{R}$, because for any $\mathbf{Z}_i \in \mathcal{Z}$, we have $k(\mathbf{Z}_i, \mathbf{Z}_j) = \langle \varphi(\mathbf{Z}_i), \varphi(\mathbf{Z}_j) \rangle_\mathcal{H}$. Formally, we have:

**Definition 2.1** (Reproducing kernel Hilbert space). Let $\mathcal{H}$ be a Hilbert space of $\mathbb{R}$-valued functions defined on a non-empty set $\mathcal{Z}$. A function $k : \mathcal{Z} \times \mathcal{Z} \to \mathbb{R}$ is called a reproducing kernel of $\mathcal{H}$, and $\mathcal{H}$ is a reproducing kernel Hilbert space, if $k$ satisfies

- $\forall \mathbf{Z}_i \in \mathcal{Z}, k(\cdot, \mathbf{Z}_i) \in \mathcal{H}$,
- $\forall \mathbf{Z}_i \in \mathcal{Z}, \forall h \in \mathcal{H}, \langle h, k(\cdot, \mathbf{Z}_i) \rangle_\mathcal{H} = h(\mathbf{Z}_i)$.

We focus on the translation-invariant kernel in our paper, where the kernel $k(\mathbf{Z}_i, \mathbf{Z}_j)$ can always be written as $k'(\mathbf{Z}_i - \mathbf{Z}_j)$ for $k' \in \mathcal{Z} \to \mathbb{R}$. The Moore–Aronszajn's theorem states that if $k$ is a symmetric, positive definite kernel on $\mathcal{Z}$, there is a unique Hilbert space of functions $\mathcal{H}$ on $\mathcal{Z}$ for which $k$ is a reproducing kernel.

For instance, the Gaussian kernel is a symmetric, positive definite kernel that yields an RKHS with infinite dimensions. One of the advantages of a reproducing kernel is that the similarity can be computed directly in $\mathcal{Z}$ without using the feature map to go to the potentially infinite dimensional Hilbert space. However, a reproducing kernel's similarity structure should ideally align with the semantic meanings of specific tasks. For example, it is unlikely to calculate the semantic similarity of two images directly using a predefined reproducing kernel in the pixel space.

Consequently, we ask if it is possible to find an embedding function $\boldsymbol{f} : \mathcal{X} \to \mathcal{Z}$, where $\mathcal{Z}$ can compute the similarity of two objects in $\mathcal{X}$ with a predefined kernel function, i.e., whether $\mathbf{K_Z}$ matches with $\boldsymbol{\pi}$ in Figure 1. In other words, we hope to map the objects to a space where the semantic similarity in $\mathcal{X}$ is naturally embedded. This is the starting point of our paper.

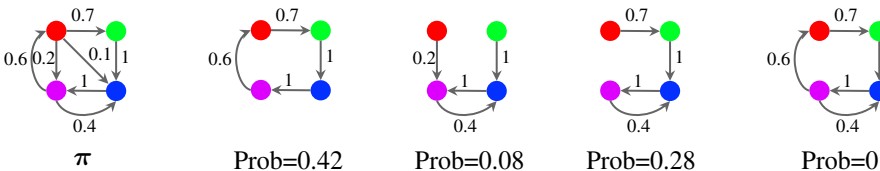

Figure 2: Sampling probabilities of the subgraphs defined by $\mathbb{P}(\mathbf{W}; \boldsymbol{\pi})$. The first subfigure represents the underlying graph $\boldsymbol{\pi}$, the next three subfigures represent three different subgraphs with their sampling probabilities. The last subfigure has sampling probability $0$ because the purple node has out-degree larger than $1$.

### 2.4 MARKOV RANDOM FIELD

In this subsection, we present the framework (without proofs) of MRF for dimension reduction (Van Assel et al., 2022). We have modified some definitions and lemmas for our learning scenarios, and the readers may check the paper for more details on this framework.

Consider $n$ objects $\mathbf{Z} = [\mathbf{Z}_1, \cdots, \mathbf{Z}_n]$ in $\mathcal{Z}$. We use a symmetric and translation invariant kernel $k : \mathcal{Z} \to \mathbb{R}_+$ to represent the similarities in $\mathcal{Z}$, where symmetric means $k(\mathbf{x}) = k(-\mathbf{x})$. Given $\mathbf{Z}$ and $k$, we define the gram matrix as $\mathbf{K}_{\mathbf{Z}} \triangleq (k(\mathbf{Z}_i - \mathbf{Z}_j))_{(i,j) \in [n]^2}$, which is also the adjacency matrix representing the similarities of objects in $\mathbf{Z}$.

Due to the large size of $\boldsymbol{\pi}$ and in practice $\boldsymbol{\pi}$ is usually formed by using positive samples sampling and hard to explicitly construct, directly comparing $\mathbf{K}_{\mathbf{Z}}$ and $\boldsymbol{\pi}$ can be difficult, so we treat them as MRFs and compare the induced probability distributions on subgraphs instead. In our paper, subgraphs are directed unweighted graphs from the set $S_{\mathbf{W}} \triangleq \{\mathbf{W} \in \{0,1\}^{n \times n} \mid \forall (i,j) \in [n]^2, \mathbf{W}_{i,i} = 0\}$. The distribution of $\mathbf{W}$ is generally defined as follows.

**Definition 2.2** (Distribution of $\mathbf{W}$). Let $\boldsymbol{\pi} \in \mathbb{R}_+^{n \times n}$, we define the distribution $\mathbb{P}(\mathbf{W}; \boldsymbol{\pi}) \propto \Omega(\mathbf{W}) \Pi_{(i,j) \in [n]^2} \boldsymbol{\pi}_{i,j}^{\mathbf{W}_{i,j}}$, where $\Omega(\mathbf{W}) \triangleq \Pi_i \mathbb{I}_{\sum_j \mathbf{W}_{i,j} = 1}$ is called a unitary out-degree filter.

To provide a clearer interpretation of the definition, we can break down the expression $\Omega(\mathbf{W}) \Pi_{(i,j) \in [n]^2} \boldsymbol{\pi}_{i,j}^{\mathbf{W}_{i,j}}$ into two parts. Firstly, if we view $\mathbf{W}$ as the adjacency of a graph, the unitary out-degree filter $\Omega(\mathbf{W})$ checks if each node $i$ of the graph has exactly one out-going edge. Therefore, only subgraphs with a unitary out-degree will be preserved, while subgraphs with other out-degree values will be filtered out. As we will see later, this exactly corresponds to the setting that the InfoNCE loss uses exactly one positive neighbor. Secondly, $\Pi_{(i,j) \in [n]^2} \boldsymbol{\pi}_{i,j}^{\mathbf{W}_{i,j}}$ multiplies the scores of each edge in $\boldsymbol{\pi}$ compared with $\mathbf{W}_{i,j}$. This multiplication results in the un-normalized likelihood of $\mathbf{W}$ under $\boldsymbol{\pi}$, it is noticeable that the constraint of single outgoing edge of each node that ensures that the multiplication results reflect the consistent of $\mathbf{W}$ and $\boldsymbol{\pi}$. See Figure 2 for an illustration.

By applying Definition 2.2 to $\mathbf{K}_{\mathbf{Z}}$, we obtain the following expression for $\mathbb{P}(\mathbf{W}; \mathbf{K}_{\mathbf{Z}})$: $\mathbb{P}(\mathbf{W}; \mathbf{K}_{\mathbf{Z}}) \propto \Omega(\mathbf{W}) \Pi_{(i,j) \in [n]^2} k(\mathbf{Z}_i - \mathbf{Z}_j)^{\mathbf{W}_{i,j}}$. This expression represents the prior probability of $\mathbf{W}$ under $\mathbf{K}_{\mathbf{Z}}$.

Due to the unitary out-degree filter, $\mathbb{P}(\mathbf{W}; \boldsymbol{\pi})$ has the following property.

**Lemma 2.3.** *For $\mathbf{W} \sim \mathbb{P}(\cdot; \boldsymbol{\pi})$, $\forall i \in [n]$, $\mathbf{W}_i \sim \mathcal{M}(1, \boldsymbol{\pi}_i / \sum_j \boldsymbol{\pi}_{i,j})$, where $\mathcal{M}$ is the multinomial distribution. Moreover, given any $i, i' \in [n]$, $\mathbf{W}_i$ is independent to $\mathbf{W}_{i'}$. Where $\mathbf{W}_i$ is the $i$-th row of $\mathbf{W}$, $\boldsymbol{\pi}_i$ is the $i$-th row of $\boldsymbol{\pi}$.*

Below we define the cross entropy loss given distribution $\boldsymbol{\pi}$ and the similarity matrix $\mathbf{K}_{\mathbf{Z}}$.

$$\mathcal{H}_{\boldsymbol{\pi}}^k(\mathbf{Z}) \triangleq -\mathbb{E}_{\mathbf{W}_{\mathbf{X}} \sim \mathbb{P}(\cdot; \boldsymbol{\pi})}[\log \mathbb{P}(\mathbf{W}_{\mathbf{Z}} = \mathbf{W}_{\mathbf{X}}; \mathbf{K}_{\mathbf{Z}})] \tag{3}$$

The following lemma will be helpful in analyzing the cross-entropy loss, which states that when the two distributions can be aligned and decomposed, their cross-entropy loss can also be decomposed.

**Lemma 2.4.** *Assume $\mathcal{X} = \mathcal{X}_1 \times \cdots \times \mathcal{X}_k$ and there are two probability distributions $\mathbf{P}$ and $\mathbf{Q}$ supported on $\mathcal{X}$. Suppose $\mathbf{P} = \mathbf{P}_1 \otimes \cdots \otimes \mathbf{P}_k$ and $\mathbf{Q} = \mathbf{Q}_1 \otimes \cdots \otimes \mathbf{Q}_k$, with $\mathbf{P}_i$ and $\mathbf{Q}_i$ supported on $\mathcal{X}_i$. Let $\mathcal{H}(\mathbf{P}, \mathbf{Q}) \triangleq -\mathbb{E}_{x \sim \mathbf{P}}[\log \mathbf{Q}(x)]$. Then $\mathcal{H}(\mathbf{P}, \mathbf{Q}) = \sum_{i=1}^k \mathcal{H}(\mathbf{P}_i, \mathbf{Q}_i)$.*

The next lemma shows that the cross-entropy loss can be converted to the combination of repulsion and attraction terms.

**Lemma 2.5.** $\min_{\mathbf{Z}} \mathcal{H}_{\boldsymbol{\pi}}^k(\mathbf{Z})$ *is equivalent to*

$$\min_{\mathbf{Z}} - \sum_{(i,j) \in [n]^2} \mathbf{P}_{i,j} \log k(\mathbf{Z}_i - \mathbf{Z}_j) + \log \mathbf{R}(\mathbf{Z}), \tag{4}$$

*where* $\mathbf{P} = \mathbb{E}_{\mathbf{W_X} \sim \mathbb{P}(\cdot;\boldsymbol{\pi})}[\mathbf{W_X}]$, *and* $\mathbf{R}(\mathbf{Z}) = \sum_{\mathbf{W} \in S_{\mathbf{W}}} \mathbb{P}(\mathbf{Z}, \mathbf{W})$ *with* $\mathbb{P}(\mathbf{Z}, \mathbf{W}) \propto f_k(\mathbf{Z}, \mathbf{W})\Omega(\mathbf{W})$.

The second term in Eqn. (4) punishes trivial solutions like $\mathbf{Z} = \mathbf{0}$, as $\mathbf{0}$ is a mode for $f_k(\cdot, \mathbf{W})$ for any $\mathbf{W}$, which incurs large $\log \mathbf{R}(\mathbf{Z})$. The first term can be interpreted using the graph Laplacian operator, defined below.

**Definition 2.6** (Graph Laplacian operator). The graph Laplacian operator is a function $\mathbf{L}$ that maps a $n \times n$ non-negative matrix to a positive semi-definite matrix such that:

$$\forall i, j \in [n]^2, \mathbf{L}(\mathbf{W})_{i,j} = \begin{cases} -\mathbf{W}_{i,j} & \text{if } i \neq j \\ \sum_{k \in [n]} \mathbf{W}_{i,k} & \text{o.w.} \end{cases}$$

We will then introduce the definition of spectral clustering used in our paper.

**Definition 2.7** (Spectral Clustering). Let $\mathbf{W} \in \mathbb{R}_+^{n \times n}$ be the adjacency matrix of a graph and $\mathbf{L}$ be the graph Laplacian operator. Then the following optimization problem is called performing spectral clustering on the graph whose adjacency matrix is $\mathbf{W}$:

$$\min_{\mathbf{Z}} \operatorname{tr}(\mathbf{Z}^\top \mathbf{L}(\mathbf{W})\mathbf{Z}) + \mathrm{E}(\mathbf{Z}), \tag{5}$$

where $\mathrm{E}(\mathbf{Z})$ is a regularization term.

By simple calculation, when $k$ is the Gaussian kernel, the first term in Eqn. (4) becomes $\operatorname{tr}(\mathbf{Z}^\top \mathbf{L}^* \mathbf{Z})$ where $\mathbf{L}^* = \mathbb{E}_{\mathbf{W_X} \sim \mathbb{P}(\cdot;\boldsymbol{\pi})}[\mathbf{L}(\mathbf{W_X})]$. In other words, Eqn. (4) is equivalent to doing spectral clustering with a repulsion regularizer $\log \mathbf{R}(\mathbf{Z})$.

## 3 CONTRASTIVE LEARNING: SIMCLR

In this section, we will prove our main theorem that contrastive learning is spectral clustering on a similarity graph. We assume that there are finitely many objects in $\mathcal{X}$, denoted as $n$. This is the same assumption used by HaoChen et al. (2021), who also demonstrated that the finite case can be easily extended to the infinite case by replacing sum by integral, adjacency matrix by adjacency operator, etc. For continuous augmentation methods like adding Gaussian noise, we can discretize it in a natural way. Assuming a finite number of objects can help us avoid non-essential technical jargon.

With $n$ objects in $\mathcal{X}$, consider a similarity graph defined on these objects, which gives a similarity matrix $\boldsymbol{\pi}$ of size $n \times n$. However, for real scenarios like learning images, it is extremely difficult to obtain such $\boldsymbol{\pi}$ from human labeling. Therefore, we compute $\boldsymbol{\pi}$ using the prior knowledge of the dataset. For example, in the original SimCLR paper (Chen et al., 2020a), there are 9 different augmentation methods. Each augmentation method may generate many different augmented images that look similar to the original image. For every original image $\mathbf{X}_i$, we define a probability distribution $\boldsymbol{\pi}_i$, such that each object $\mathbf{X}_j$ gets sampled with probability $\boldsymbol{\pi}_{i,j}$. For example, during the augmentation process, suppose $\mathbf{X}_j$ has a probability, say $1/9$, to be an augmentation of $\mathbf{X}_i$, then $\boldsymbol{\pi}_{i,j} = 1/9$. Therefore, $\boldsymbol{\pi}_i$ can be represented as a vector in $\mathbb{R}_+^n$.

Stacking all probability distributions $\boldsymbol{\pi}_i$ together for $i \in [n]$, we get a matrix $\boldsymbol{\pi} \in \mathbb{R}_+^{n \times n}$. In this section, we assume the sampling process is symmetric, i.e., $\boldsymbol{\pi}_{i,j} = \boldsymbol{\pi}_{j,i}$. The stochastic data augmentation samples $\mathbf{W_X}$ based on $\boldsymbol{\pi}$, i.e., $\mathbf{W_X} \sim \mathbb{P}(\cdot; \boldsymbol{\pi})$.

## 3.1 MAIN THEOREM

**Theorem 3.1.** *For the SimCLR algorithm, denote $f$ as the neural network, $\mathbf{Z} := f(\mathbf{X})$, and $\boldsymbol{\pi}$ as the similarity graph defined by the data augmentation process where objects are connected iff they are positive samples of each other. Then SimCLR is equivalent to solving the following program:*

$$\min_{\mathbf{Z}} \operatorname{tr}(\mathbf{Z}^\top \mathbf{L}(\boldsymbol{\pi})\mathbf{Z}) + \log \mathbf{R}(\mathbf{Z}),$$

*which runs spectral clustering on $\boldsymbol{\pi}$.*

*Proof.* Please refer to Appendix A. □

**Discussions.** Empirically, the InfoNCE loss is applied to a large batch of the object, rather than all the $n$ objects that Theorem 3.1 requires. This explains why SimCLR benefits from larger batch size, e.g. Chen et al. (2020a) use a batch size of 4096, and He et al. (2019b) use an even large memory bank for storing more samples.

While using the same framework from (Van Assel et al., 2022), our Theorem 3.1 is significantly different from their results on dimension reduction from at least two aspects. Firstly, in the object space $\mathcal{X}$, we have a predefined similarity graph $\boldsymbol{\pi}$, but they were using a kernel matrix $\mathbf{K_X}$ based on $\mathbf{X}$ and a kernel $k_{\mathcal{X}}$. This is because in the dimension reduction setting, the input objects are assumed to be well-structured data points, but in the self-supervised learning setting, the input objects are images or texts, where a translation invariant kernel cannot be used for computing the similarities. Secondly, their cross-entropy loss is directly computed from $\mathbf{K_X}$, while $\mathbf{W_X}$ is never explicitly sampled. In contrast, in our method, $\boldsymbol{\pi}$ is never explicitly used, and the cross entropy loss is indirectly computed from the randomly sampled $\mathbf{W_X}$.

The equivalence we proved is exact. Therefore, after learning, the embedding space contains different components, corresponding to various (sub-) classes of the objects. This characterization naturally explains why contrastive learning works well for classification-related downstream tasks.

## 4 MULTI-MODAL LEARNING: CLIP

In this subsection, we extend Theorem 3.1 to the multi-modal setting by analyzing CLIP, which applies the contrastive loss to the image-text pairs. The image-text pairs can be represented with the following pair graph.

**Definition 4.1** (Pair graph). Consider two modalities of objects $\mathbf{A}, \mathbf{B}$, and undirected unit-weight edges $\mathbf{E} = \{(\mathbf{a}_i, \mathbf{b}_i) \mid \mathbf{a}_i \in \mathbf{A}, \mathbf{b}_i \in \mathbf{B}\}_{i=1}^{M}$. The pair graph between $\mathbf{A}, \mathbf{B}$ is a directed bipartite graph $\boldsymbol{\pi}_{\mathbf{A},\mathbf{B}} = (\mathbf{A}, \mathbf{B}, \mathbf{E})$, with the weight of each outgoing edge normalized by the out-degree of the node.

By definition, $\boldsymbol{\pi}_{\mathbf{A},\mathbf{B}}$ is not necessarily symmetric. Consider the case where the dataset contains 10 images of "dog", all of them are connected to the same text "dog". In this case, the text dog has $1/10$ probability to each image, while each image has only one edge with $100\%$ probability to the text. However, since each row of $\boldsymbol{\pi}_{\mathbf{A},\mathbf{B}}$ is still a probability distribution, we still have the next theorem.

**Theorem 4.2** (CLIP's objective). *For the CLIP algorithm, denote $\boldsymbol{\pi}_{\mathbf{A},\mathbf{B}}$ as the pair graph. Then CLIP is equivalent to running the generalized spectral clustering on $\boldsymbol{\pi}_{\mathbf{A},\mathbf{B}}$.*

*Proof.* Please refer to Appendix A. □

**Discussions.** In Theorem 4.2, we say CLIP runs the generalized spectral clustering because $\mathbf{L}(\boldsymbol{\pi}_{\mathbf{A},\mathbf{B}})$ is not necessarily the Laplacian of a symmetric graph, although one can still compute the optimal embedding $\mathbf{Z}$ following Eqn. (4). The pair graph may contain a huge number of isolated edges. Empirically, CLIP picks strong image and text encoders with good prior knowledge about the dataset. Such prior knowledge may bias towards a better embedding for grouping the isolated edges with more semantics.

Theorem 4.2 also assumes that all the objects are sampled in $\mathbf{W_X}$, while empirically a really big batch size of 32,768 is used in Radford et al. (2021). Moreover, the probability distribution $\mathbb{P}(\cdot; \boldsymbol{\pi}_{\mathbf{A},\mathbf{B}})$

used in Theorem 4.2 is slightly different from the implementation of CLIP, in the sense that CLIP uniformly samples the edges in $\mathbf{E}$, but here we uniformly sample the objects in $\mathbf{A} \cup \mathbf{B}$. When the image-text pairs dataset has high quality, the difference between these two sampling schemes becomes negligible as the variance of object out-degrees is extremely small.

## 4.1 APPLYING TO LACLIP

Due to computation resources limitations, we haven't implemented an improved CLIP algorithm ourselves. Interestingly, a direct improvement to CLIP based on our theory was recently conducted. We shall present this algorithm LaCLIP carefully (Fan et al., 2023) and discuss why it can be seen as a direct application of our theory.

Roughly speaking, LaCLIP is a direct extension of CLIP by not only incorporating image augmentations but using text augmentation as well. Specifically, we can treat language rewrites as text augmentations. For each image text pair $(x_I, x_T)$, the text augmentation can be derived as follows:

$$\text{aug}_T(x_T) \sim \text{Uniform}([x_{T_0}, x_{T_1} \ldots, x_{T_M}]), \tag{6}$$

where $x_{T_i}$ is the text $x_T$ itself or its rewrite.

Then training loss over the images in LaCLIP becomes:

$$\mathcal{L}_I := -\sum_{i=1}^{N} \log \frac{\exp\left(\text{sim}\left(\boldsymbol{f}_I\left(\text{aug}_I\left(x_I^i\right)\right), \boldsymbol{f}_T\left(\text{aug}_T\left(x_T^i\right)\right)\right)/\tau\right)}{\sum_{k=1}^{N} \exp\left(\text{sim}\left(\boldsymbol{f}_I\left(\text{aug}_I\left(x_I^i\right)\right), \boldsymbol{f}_T\left(\text{aug}_T\left(x_T^k\right)\right)\right)/\tau\right)},$$

where $\boldsymbol{f}_I$ and $\boldsymbol{f}_T$ are image and text encoders respectively.

From the pair graph point of view, LaCLIP expands the nodes in the "text side" of the pair graph by including the nodes of the rewrite (augmented) texts. Moreover, as the augmented images are connected to augmented texts, this augmented pair graph will have more clusters between similar objects than the original CLIP pair graph. Thus, from the spectral clustering, it will be natural to expect LaCLIP shall cluster similar objects across modalities better than CLIP. Indeed, the zero-shot transfer ability of LaCLIP significantly improves (Fan et al., 2023).

## 5 USING NEW KERNELS

### 5.1 MAXIMUM ENTROPY PRINCIPLE

In this subsection, we offer an interpretation of InfoNCE-like loss, suggesting that exponential kernels are natural choices to use in this type of loss. Given a query sample $\mathbf{q}$, let $\psi_i$ represent the similarity between $\mathbf{q}$ and the contrastive sample $\mathbf{p}_i$ for $i \in [n]$, computed by a kernel $k$. Without loss of generality, assume $\mathbf{p}_1$ is the (positive) neighbor of $\mathbf{q}$ according to prior knowledge, but $\psi_1$ is not necessarily the largest value in $\psi_i$. Ideally, we desire $\psi_1$ to be the largest or at least among the few largest similarities, which indicates that our kernel properly aligns with the prior knowledge of $\mathcal{X}$.

To optimize toward this goal, we must design a loss function that captures the ranking of $\psi_1$. Since the ordering function is discrete and lacks gradient information, we need to convert it into a soft and continuous function that enables gradient-based optimization. Specifically, we employ a probability distribution $\boldsymbol{\alpha}$ to represent the neighborhood structure of $\mathbf{q}$ in relation to $\psi_1$, satisfying $\psi_1 \leq \sum_{i=1}^{n} \alpha_i \psi_i$, and $\forall i, \alpha_i \geq 0$. If $\psi_1$ is the largest, $\boldsymbol{\alpha} = e_1$ is the sole solution; otherwise, $\boldsymbol{\alpha}$ can be more diverse. For instance, when all $\psi_i$ values are equal, $\boldsymbol{\alpha}$ can be a uniform distribution.

Intuitively, if there are numerous other $\psi_i$ values similar to $\psi_1$, the neighborhood structure of $\mathbf{q}$ is not as optimal as when $\psi_1$ is the only object close to $\mathbf{q}$. Formally, this means $\boldsymbol{\alpha}$ should have fewer non-zero entries or at least concentrate on $\boldsymbol{\alpha}_1$. We use its entropy $H(\boldsymbol{\alpha}) = -\sum_{i=1}^{n} \alpha_i \log \alpha_i$ to represent this diversity, which results in the following optimization problem.

$$
\begin{aligned}
(\text{P1}) \quad \max_{\boldsymbol{\alpha}} \quad & H(\boldsymbol{\alpha}) \\
\text{s.t.} \quad & \boldsymbol{\alpha}^\top \mathbf{1}_n = 1, \alpha_1, \ldots, \alpha_n \geq 0 \\
& \psi_1 - \sum_{i=1}^{n} \alpha_i \psi_i \leq 0
\end{aligned}
$$

By minimizing the solution of (P1), we can discover an embedding that more accurately approximates the prior knowledge. However, how can we solve (P1)? By introducing the Lagrangian dual variable

$\tau > 0$, we obtain the subsequent program (P2)'s solution upper bound $\tau$ times the solution of (P1). Consequently, minimizing (P2) simultaneously produces a smaller upper bound of (P1) as well, indirectly aiding us in achieving our objective.

$$\text{(P2)} \quad \max_{\boldsymbol{\alpha}} \quad -E(\boldsymbol{\alpha})$$
$$\text{s.t.} \quad \boldsymbol{\alpha}^\top \mathbf{1}_n = 1, \alpha_1, \ldots, \alpha_n \geq 0$$

where $E(\boldsymbol{\alpha}) = \psi_1 - \sum_{i=1}^n \alpha_i \psi_i + \tau \sum_{i=1}^n \alpha_i \log \alpha_i$.

We present the following theorem for solving (P2).

**Theorem 5.1** (Exponential kernels are natural). *The solution of (P2) satisfies:*

$$-E(\boldsymbol{\alpha}^*) = -\tau \log \frac{\exp\left(\frac{1}{\tau}\psi_1\right)}{\sum_{i=1}^n \exp\left(\frac{1}{\tau}\psi_i\right)}.$$

*Proof.* Please refer to Appendix A. $\qquad\square$

Using this framework, we can derive results akin to the max-min optimization formulation from Tian (2022) as a corollary.

## 5.2 KERNEL-INFONCE LOSS

In this subsection, we will show how to use the derivation above to improve InfoNCE loss. Theorem 5.1 suggests that the loss function of the form $-\tau \log \frac{\exp\left(\frac{1}{\tau}\psi_1\right)}{\sum_{i=1}^n \exp\left(\frac{1}{\tau}\psi_i\right)}$ is a natural choice for characterizing the neighborhood similarity structure. When the similarity between the query sample $\mathbf{p}$ and neighbourhood sample $\mathbf{p}_i$ is defined as $\psi_i = C - \|f(\mathbf{q}) - f(\mathbf{p}_i)\|^\gamma$, where $C$ is a large positive constant and $\gamma > 0$. We find it recovers the exponential kernels defined as follows:

$$K_{\exp}^{\gamma,\tau}(\mathbf{x}, \mathbf{y}) := \exp\left(-\frac{\|\mathbf{x} - \mathbf{y}\|^\gamma}{\tau}\right) \quad (\gamma, \tau > 0) \tag{7}$$

We then define our kernel-based contrastive loss, Kernel-InfoNCE, as follows:

$$\mathcal{L}_{\text{Kernel-InfoNCE}}^{\gamma,\tau}(\mathbf{q}, \mathbf{p}_1, \{\mathbf{p}_i\}_{i=2}^N) := -\log \frac{K_{\exp}^{\gamma,\tau}(\mathbf{q}, \mathbf{p}_1)}{\sum_{i=1}^N K_{\exp}^{\gamma,\tau}(\mathbf{q}, \mathbf{p}_i)}. \tag{8}$$

Note equation (8) can be easily derived by setting the kernel $k$ in equation (3) to exponential kernel. Our framework in Section 3 is suitable for explaining losses that are adapted from InfoNCE by changing kernels. We consider generalizing the exponential kernel a bit. We propose to use the mixture of two exponential kernels as potential candidates for replacing the Gaussian kernel. There are two kinds of mixing methods. The first one is taking the weighted average of two positive definite kernels.

The other mixing method is concatenation, which splits the input vectors into two parts, where the first part uses the first kernel, and the second part uses the second kernel. It is easy to see that both mixing methods maintain the strictly positive definite property of the base kernels. We list the two types of kernel mixtures below.

**Simple Sum Kernel:**

$$K(x_i, x_j) := \exp(-\|\boldsymbol{f}(\mathbf{x}_i) - \boldsymbol{f}(\mathbf{x}_j)\|_2^2 / \tau_2) + \exp(-\|\boldsymbol{f}(\mathbf{x}_i) - \boldsymbol{f}(\mathbf{x}_j)\|_2^1 / \tau_1)$$

**Concatenation Sum Kernel:**

$$K(x_i, x_j) := \exp(-\|\boldsymbol{f}(\mathbf{x}_i)[0:n] - \boldsymbol{f}(\mathbf{x}_j)[0:n]\|_2^2 / \tau_2) + \exp(-\|\boldsymbol{f}(\mathbf{x}_i)[n:2n] - \boldsymbol{f}(\mathbf{x}_j)[n:2n]\|_2^1 / \tau_1)$$

## 6 EXPERIMENTS

In our experiments, we reproduce the baseline algorithm SimCLR (Chen et al., 2020a), and replace SimCLR's Gaussian kernel with other kernels. We then test against SimCLR using Kernel-InfoNCE

Table 1: Results on CIFAR-10, CIFAR-100, and TinyImageNet datasets.

| Method | CIFAR-10 | | CIFAR-100 | | TinyImageNet | |
|---|---|---|---|---|---|---|
| | 200 epochs | 400 epochs | 200 epochs | 400 epochs | 200 epochs | 400 epochs |
| SimCLR (repro.) | $88.11 \pm 0.09$ | $90.60 \pm 0.15$ | $62.57 \pm 0.10$ | $66.29 \pm 0.12$ | $34.00 \pm 0.18$ | $37.83 \pm 0.09$ |
| Laplacian Kernel | $89.26 \pm 0.18$ | $91.03 \pm 0.17$ | $63.16 \pm 0.15$ | $66.09 \pm 0.11$ | $35.91 \pm 0.21$ | $38.71 \pm 0.18$ |
| $\gamma = 0.5$ Exponential Kernel | $89.00 \pm 0.07$ | $91.23 \pm 0.12$ | $63.48 \pm 0.22$ | $65.81 \pm 0.19$ | $34.17 \pm 0.13$ | $38.75 \pm 0.15$ |
| **Simple Sum Kernel** | $89.82 \pm 0.09$ | $\mathbf{91.72 \pm 0.10}$ | $\mathbf{66.67 \pm 0.20}$ | $\mathbf{68.62 \pm 0.15}$ | $\mathbf{36.61 \pm 0.13}$ | $\mathbf{39.38 \pm 0.16}$ |
| Concatenation Sum Kernel | $\mathbf{89.89 \pm 0.18}$ | $91.28 \pm 0.07$ | $66.10 \pm 0.21$ | $68.57 \pm 0.11$ | $35.98 \pm 0.17$ | $38.77 \pm 0.22$ |

loss on various benchmark vision datasets, including CIFAR-10/100 (Krizhevsky et al., 2009) and TinyImageNet (Le & Yang, 2015).

For each algorithm, we first train an encoder $f$ on the training dataset to minimize the empirical loss function generated by the kernel. Then, following the standard linear evaluation protocol, we freeze the encoder $f$ and train a supervised linear classifier, which takes the output representation of $f$ as input. Additional experimental details, dataset information, and results can be found in Appendix B.

**Experimental Results.** We summarize our empirical results on various benchmark datasets in Table 1. It is evident that we have achieved better performance than SimCLR on all three benchmark datasets, with the Simple Sum Kernel reaching the best average performance.

# 7 RELATED WORK

Contrastive learning constitutes a classical method extensively employed in representation learning (Hadsell et al., 2006; Becker & Hinton, 1992). Owing to its recent applications in self-supervised learning, contrastive learning has garnered widespread attention and achieved state-of-the-art results in numerous downstream tasks within computer vision (Tian et al., 2020; Cui et al., 2021), graph representation learning (You et al., 2020; Hassani & Khasahmadi, 2020; Deng et al., 2022), multi-modality (Radford et al., 2021), and beyond. Contrastive predictive coding (Oord et al., 2018) represents one of the pioneering methods to incorporate the concept of contrastive learning in self-supervised learning. Subsequently, various methods have sought to enhance performance. SimCLR (Chen et al., 2020a) and MoCo (Chen et al., 2020b) advocate for utilizing a large batch size and momentum update mechanism to guarantee effective learning. Moreover, the hard negative sampling method (Kalantidis et al., 2020) has been explored to mitigate the impact of false negative sampling.

Despite the empirical success of contrastive learning, the theoretical comprehension of its underlying mechanisms remains limited. Oord et al. (2018) demonstrate that the InfoNCE loss can be regarded as a surrogate loss for maximizing mutual information. Arora et al. (2019) give the generalization bound for contrastive learning under a latent class assumption. HaoChen et al. (2021) incorporate the concept of augmentation graph to facilitate the analysis of contrastive learning and propose a surrogate loss spectral contrastive loss. They show that this surrogate loss is equivalent to spectral clustering on augmentation graph. Wang et al. (2022) propose that aggressive data augmentations lead to overlapping support of intra-class samples, allowing for the clustering of positive samples and the gradual learning of class-separated representations, providing new insights of understanding contrastive learning. Balestriero & LeCun (2022) link a variant of contrastive loss to the ISOMAP algorithm. Hu et al. (2022) connect contrastive learning with stochastic neighbor embedding. Wang & Isola (2020) reveal that the quality of embedding can be decomposed into an alignment component and a uniformity component, considering both the loss function and the embedding space.

# 8 CONCLUSION

In this paper, we take the probabilistic perspective of contrastive learning, and prove that it is essentially running spectral clustering on the predefined similarity graph. Extending this result to multi-modal learning, we show that CLIP is also doing the generalized spectral clustering on the pair graph. Based on the maximum entropy principle and other useful properties, we propose to use the mixtures of exponential kernels (Kernel-InfoNCE loss) to replace the Gaussian kernel, which has achieved better performance empirically.

## REPRODUCIBILITY STATEMENT

For reproducibility, we share our code at `https://github.com/yifanzhang-pro/Kernel-InfoNCE`. The experiment results can be reproduced following the instructions in the README document. We also provide our experiment details in Appendix B.

## ACKNOWLEDGMENTS

The authors would like to thank Van Assel for clarifying a derivation step in his paper, and anonymous reviewers and ACs for their helpful suggestions. This work is supported by the Ministry of Science and Technology of the People's Republic of China, the 2030 Innovation Megaprojects "Program on New Generation Artificial Intelligence" (Grant No. 2021AAA0150000).

## AUTHOR CONTRIBUTIONS

Yifan Zhang suggests the relationship between MRF and contrastive learning. Zhiquan Tan discovered the MRF framework for dimension reduction (Van Assel et al., 2022), and applied this framework to prove Theorem 3.1. He also introduced the viewpoint of the maximum entropy principle for contrastive learning. Yifan and Zhiquan proposed Kernel-InfoNCE loss and refined the paper. Jingqin Yang did comprehensive experiments on Kernel-InfoNCE loss. Yang Yuan extended Theorem 3.1 to CLIP, and wrote most of the paper.

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

## A  APPENDIX FOR PROOFS

**Proof of Theorem 3.1.**

*Proof.* Our proof has two steps. In Step 1, we will show that SimCLR is equivalent to minimizing the cross entropy loss defined in Eqn. (3). In Step 2, we will show that minimizing the cross-entropy loss is equivalent to spectral clustering on $\boldsymbol{\pi}$. Combining the two steps together, we have proved our theorem.

**Step 1:** SimCLR is equivalent to minimizing the cross entropy loss.

The cross entropy loss takes expectation over $\mathbf{W_X} \sim \mathbb{P}(\cdot; \boldsymbol{\pi})$, which means $\mathbf{W_X}$ has exactly one non-zero entry in each row $i$. By Lemma 2.3, we know every row $i$ of $\mathbf{W_X}$ is independent of other rows. Moreover, $\mathbf{W}_{\mathbf{X},i} \sim \mathcal{M}(1, \boldsymbol{\pi}_i / \sum_j \boldsymbol{\pi}_{i,j}) = \mathcal{M}(1, \boldsymbol{\pi}_i)$, because $\boldsymbol{\pi}_i$ itself is a probability distribution. Similarly, we know $\mathbf{W_Z}$ also has the row-independent property by sampling over $\mathbb{P}(\cdot; \mathbf{K_Z})$. Therefore, by Lemma 2.4, we know Eqn. (3) is equivalent to:

$$-\sum_{i=1}^n \mathbb{E}_{\mathbf{W}_{\mathbf{X},i}}[\log \mathbb{P}(\mathbf{W}_{\mathbf{Z},i} = \mathbf{W}_{\mathbf{X},i}; \mathbf{K_Z})],$$

This expression takes expectation over $\mathbf{W}_{\mathbf{X},i}$ for the given row $i$. Notice that $\mathbf{W}_{\mathbf{X},i}$ has exactly one non-zero entry, which equals 1 (same for $\mathbf{W}_{\mathbf{Z},i}$). As a result we expand the above expression to be:

$$-\sum_{i=1}^n \sum_{j \neq i} \Pr(\mathbf{W}_{\mathbf{X},i,j} = 1) \log \Pr(\mathbf{W}_{\mathbf{Z},i,j} = 1). \tag{9}$$

By Lemma 2.3, $\Pr(\mathbf{W}_{\mathbf{Z},i,j} = 1) = \mathbf{K}_{\mathbf{Z},i,j} / \|\mathbf{K}_{\mathbf{Z},i}\|_1$ for $j \neq i$. Recall that $\mathbf{K_Z} = (k(\mathbf{Z}_i - \mathbf{Z}_j))_{(i,j) \in [n]^2}$, which means $\mathbf{K}_{\mathbf{Z},i,j} / \|\mathbf{K}_{\mathbf{Z},i}\|_1 = \frac{\exp(-\|\mathbf{Z}_i - \mathbf{Z}_j\|^2 / 2\tau)}{\sum_{k \neq i} \exp(-\|\mathbf{Z}_i - \mathbf{Z}_k\|^2 / 2\tau)}$ for $j \neq i$, when $k$ is the Gaussian kernel with variance $\tau$.

Notice that $\mathbf{Z}_i = f(\mathbf{X}_i)$, so we know

$$-\log \Pr(\mathbf{W}_{\mathbf{Z},i,j} = 1) = -\log \frac{\exp(-\|f(\mathbf{X}_i) - f(\mathbf{X}_j)\|^2 / 2\tau)}{\sum_{k \neq i} \exp(-\|f(\mathbf{X}_i) - f(\mathbf{X}_k)\|^2 / 2\tau)}, \tag{10}$$

The right hand side is exactly the InfoNCE loss defined in Eqn. (1). Inserting Eqn. (10) into Eqn. (9), we get the SimCLR algorithm, which first samples augmentation pairs $(i, j)$ with $\Pr(\mathbf{W}_{\mathbf{X},i,j} = 1)$ for each row $i$, and then optimize the InfoNCE loss.

**Step 2:** minimizing the cross entropy loss is equivalent to spectral clustering on $\boldsymbol{\pi}$.

By Lemma 2.5, we may further convert the loss to

$$\min_{\mathbf{Z}} - \sum_{(i,j) \in [n]^2} \mathbf{P}_{i,j} \log k(\mathbf{Z}_i - \mathbf{Z}_j) + \log \mathbf{R}(\mathbf{Z}). \tag{11}$$

Since $k$ is the Gaussian kernel, this reduces to

$$\min_{\mathbf{Z}} \operatorname{tr}(\mathbf{Z}^\top \mathbf{L}(\boldsymbol{\pi}) \mathbf{Z}) + \log \mathbf{R}(\mathbf{Z}),$$

where we use the fact that $\mathbb{E}_{\mathbf{W_X} \sim \mathbb{P}(\cdot; \boldsymbol{\pi})}[\mathbf{L}(\mathbf{W_X})] = \mathbf{L}(\boldsymbol{\pi})$, because the Laplacian operator is linear and $\mathbb{E}_{\mathbf{W_X} \sim \mathbb{P}(\cdot; \boldsymbol{\pi})}(\mathbf{W_X}) = \boldsymbol{\pi}$.  □

**Proof of Theorem 4.2.**

*Proof.* Since $\mathbf{W_X} \sim \mathbb{P}(\cdot; \boldsymbol{\pi}_{\mathbf{A},\mathbf{B}})$, we know $\mathbf{W_X}$ has exactly one non-zero entry in each row, denoting the pair that got sampled. A notable difference compared to the previous proof is we now have $n_{\mathcal{A}} + n_{\mathcal{B}}$ objects in our graph. CLIP deals with this by taking a mini-batch of size $2N$, such that

$n_\mathcal{A} = n_\mathcal{B} = N$, and adding the $2N$ InfoNCE losses together. We label the objects in $\mathcal{A}$ as $[n_\mathcal{A}]$, and the objects in $\mathcal{B}$ as $\{n_\mathcal{A} + 1, \cdots, n_\mathcal{A} + n_\mathcal{B}\}$.

Notice that $\pi_{\mathbf{A},\mathbf{B}}$ is a bipartite graph, so the edges of objects in $\mathcal{A}$ will only connect to object in $\mathcal{B}$ and vice versa. We can define the similarity matrix in $\mathcal{Z}$ as $\mathbf{K_Z}$, where $\mathbf{K_Z}(i, j + n_\mathcal{A}) = \mathbf{K_Z}(j + n_\mathcal{A}, i) = k(\mathbf{Z}_i - \mathbf{Z}_j)$ for $i \in [n_\mathcal{A}], j \in [n_\mathcal{B}]$, and otherwise we set $\mathbf{K_Z}(i, j) = 0$. The rest is same as the previous proof. □

**Proof of Theorem 5.1.**

*Proof.* Since the objective function consists of a linear term combined with an entropy regularization, which is a strongly concave function, the maximization problem is a convex optimization problem. Owing to the implicit constraints provided by the entropy function, the problem is equivalent to having only the equality constraint. We then introduce the Lagrangian multiplier $\lambda$ and obtain the following relaxed problem:

$$\widetilde{E}(\boldsymbol{\alpha}) = \psi_1 - \sum_{i=1}^{n} \alpha_i \psi_i + \tau \sum_{i=1}^{n} \alpha_i \log \alpha_i + \lambda \left( \boldsymbol{\alpha}^\top \mathbf{1}_n - 1 \right).$$

As the relaxed problem is unconstrained, taking the derivative with respect to $\alpha_i$ yields

$$\frac{\partial \widetilde{E}(\boldsymbol{\alpha})}{\partial \alpha_i} = -\psi_i + \tau \left( \log \alpha_i + \alpha_i \frac{1}{\alpha_i} \right) + \lambda = 0.$$

Solving the above equation implies that $\alpha_i$ takes the form $\alpha_i = \exp\left(\frac{1}{\tau}\psi_i\right) \exp\left(\frac{-\lambda}{\tau} - 1\right)$. Since $\alpha_i$ lies on the probability simplex, the optimal $\alpha_i$ is explicitly given by $\alpha_i^* = \frac{\exp\left(\frac{1}{\tau}\psi_i\right)}{\sum_{i'=1}^{n} \exp\left(\frac{1}{\tau}\psi_{i'}\right)}$. Substituting the optimal point into the objective function, we obtain

$$E(\boldsymbol{\alpha}^*) = \psi_1 - \sum_{i=1}^{n} \frac{\exp\left(\frac{1}{\tau}\psi_i\right)}{\sum_{i'=1}^{n} \exp\left(\frac{1}{\tau}\psi_{i'}\right)} \psi_i + \tau \sum_{i=1}^{n} \frac{\exp\left(\frac{1}{\tau}\psi_i\right)}{\sum_{i'=1}^{n} \exp\left(\frac{1}{\tau}\psi_{i'}\right)} \log \frac{\exp\left(\frac{1}{\tau}\psi_i\right)}{\sum_{i'=1}^{n} \exp\left(\frac{1}{\tau}\psi_{i'}\right)}$$

$$= \psi_1 - \tau \log \left( \sum_{i=1}^{n} \exp\left(\frac{1}{\tau}\psi_i\right) \right).$$

Thus, the Lagrangian dual function is given by

$$-E(\boldsymbol{\alpha}^*) = -\tau \log \frac{\exp\left(\frac{1}{\tau}\psi_1\right)}{\sum_{i=1}^{n} \exp\left(\frac{1}{\tau}\psi_i\right)}. \qquad\qquad □$$

## B  MORE ON EXPERIMENT DETAILS

**CIFAR-10 and CIFAR-100.**  CIFAR-10 (Krizhevsky et al., 2009) and CIFAR-100 (Krizhevsky et al., 2009) are well-known classic image classification datasets. Both CIFAR-10 and CIFAR-100 contain a total of 60k $32 \times 32$ labeled images of different classes, with 50k for training and 10k for testing. CIFAR-10 is similar to CIFAR-100, except there are 10 different classes in CIFAR-10 and 100 classes in CIFAR-100.

**TinyImageNet.**  TinyImageNet (Le & Yang, 2015) is a subset of ImageNet (Deng et al., 2009). There are 200 different object classes in TinyImageNet, with 500 training images, 50 validation images, and 50 test images for each class. All the images in TinyImageNet are colored and labeled with a size of $64 \times 64$.

**Pseudo-code.** Algorithm 1 presents the pseudo-code for our empirical training procedure.

We also provide the pseudo-code for our core loss function used in the training procedure in Algorithm 2. The pseudo-code is almost identical to SimCLR's loss function, with the exception of an extra parameter $\gamma$.

---

**Algorithm 1** Training Procedure

---

**Require:** trainable encoder network $f$, batch size $N$, augmentation strategy *aug*, loss function $L$
    with hyperparameters *args*
  1: **for** sampled minibatch $x_{i_{i=1}^N}$ **do**
  2:    **for all** $i \in 1, ..., N$ **do**
  3:       draw two augmentations $t_i = aug\,(x_i)$, $t_i' = aug\,(x_i)$
  4:       $z_i = f\,(t_i)$, $z_i' = f\,(t_i')$
  5:    **end for**
  6:    compute loss $\mathcal{L} = L(N, z, z', args)$
  7:    update encoder network $f$ to minimize $\mathcal{L}$
  8: **end for**
  9: **Return** encoder network $f$

---

---

**Algorithm 2** Core loss function $\mathcal{C}$

---

**Require:** batch size $N$, two encoded minibatches $z_1, z_2$, $\gamma$, temperature $\tau$
  1: $z = concat\,(z_1, z_2)$
  2: **for** $i \in 1, ..., 2N, j \in 1, ..., 2N$ **do**
  3:    $s_{i,j} = \|z_i - z_j\|_2^\gamma$
  4: **end for**
  5: **define** $l(i, j)$ **as** $l(i, j) = -\log \frac{exp(s_{i,j}/\tau)}{\sum_{k=1}^{2N} \mathbf{1}[k \neq i] exp(s_{i,j}/\tau)}$
  6: **Return** $\frac{1}{2N} \sum_{k=1}^{N} [l(i, i + N) + l(i + N, i)]$

---

Utilizing the core loss function $\mathcal{C}$, we can define all kernel loss functions used in our experiments in Table 2. For all $z_i \in z$ with even dimensions $n$, we define $z_{L_i} = z_i\,[0 : n/2]$ and $z_{R_i} = z_i\,[n/2 : n]$.

| Kernel | Loss function |
|---|---|
| Laplacian | $\mathcal{C}\,(N, z, z', \gamma = 1, \tau)$ |
| Sum | $\lambda * \mathcal{C}\,(N, z, z', \gamma = 1, \tau_1) + (1 - \lambda) * \mathcal{C}\,(N, z, z', \gamma = 2, \tau_2)$ |
| Concatenation Sum | $\lambda * \mathcal{C}\,(N, z_L, z_L', \gamma = 1, \tau_1) + (1 - \lambda) * \mathcal{C}\,(N, z_R, z_R', \gamma = 2, \tau_2)$ |
| $\gamma = 0.5$ | $\mathcal{C}\,(N, z, z', \gamma = 0.5, \tau)$ |

Table 2: Definition of kernel loss functions in our experiments

**Baselines.** We reproduce the SimCLR algorithm using PyTorch Lightning (Team, 2022).

**Encoder details.** The encoder $f$ consists of a backbone network and a projection network. We employ ResNet50 (He et al., 2016) as the backbone and a 2-layer MLP (connected by a batch normalization (Ioffe & Szegedy, 2015) layer and a ReLU Nair & Hinton (2010) layer) with hidden dimensions 2048 and output dimensions 128 (or 256 in the concatenation kernel case).

**Encoder hyperparameter tuning.** For each encoder training case, we randomly sample 500 hyperparameter groups (sample details are shown in Table 3) and train these samples simultaneously using Ray Tune (Liaw et al., 2018), with the ASHA scheduler (Li et al., 2018). Ultimately, the hyperparameter group that maximizes the online validation accuracy (integrated in PyTorch Lightning) within 5000 validation steps is chosen for the given encoder training case.

**Encoder training.** We train each encoder using the LARS optimizer (You et al., 2017), LambdaLR Scheduler in PyTorch, momentum 0.9, weight decay $10^{-6}$, batch size 256, and the aforementioned hyperparameters for 400 epochs on a single A-100 GPU.

**Image transformation.** The image transformation strategy, including augmentation, is identical to the default transformation strategy provided by PyTorch Lightning.

**Linear evaluation.** The linear head is trained using the SGD optimizer with a cosine learning rate scheduler, batch size 64, and weight decay $10^{-6}$ for 100 epochs. The learning rate starts at 0.3 and ends at 0.

| Hyperparameter | Sample Range | Sample Strategy |
|---|---|---|
| start learning rate | $\left[10^{-2}, 10\right]$ | log uniform |
| $\lambda$ | $[0, 1]$ | uniform |
| $\tau, \tau_1, \tau_2$ | $[0, 1]$ | log uniform |

Table 3: Hyperparameters sample strategy

**Moco Experiments.** We also tested our method based on MoCo (He et al., 2019a). The results are summarized in Table 4. Here we choose ResNet18 (He et al., 2016) as the backbone and set a temperature of $0.1$ as default. For our simple sum kernel, we set $\lambda = 0.8$. The results show that our method outperforms the original MoCo method.

Table 4: MoCo Experiment Results on CIFAR-10 and CIFAR-100.

| Method | CIFAR-10 | | | CIFAR-100 | | |
|---|---|---|---|---|---|---|
| | 200 epochs | 400 epochs | 1000 epochs | 200 epochs | 400 epochs | 1000 epochs |
| MoCo (repro.) | $76.41 \pm 0.12$ | $80.01 \pm 0.15$ | $84.45 \pm 0.08$ | $\mathbf{47.02 \pm 0.11}$ | $52.50 \pm 0.07$ | $57.62 \pm 0.15$ |
| Laplacian Kernel | $78.09 \pm 0.10$ | $\mathbf{83.85 \pm 0.09}$ | $\mathbf{88.34 \pm 0.16}$ | $46.12 \pm 0.22$ | $53.44 \pm 0.17$ | $59.10 \pm 0.14$ |
| Simple Sum Kernel | $\mathbf{78.12 \pm 0.15}$ | $83.23 \pm 0.18$ | $87.50 \pm 0.20$ | $46.65 \pm 0.06$ | $\mathbf{53.62 \pm 0.19}$ | $\mathbf{59.83 \pm 0.12}$ |

## C  EXPERIMENTS ON SYNTHETIC DATA

Consider a scenario with $n$ clusters, each containing $k$ vertices. Let the probability of vertices $u$ and $v$ from the same cluster belonging to $\boldsymbol{\pi}$ be $p$. Conversely, for vertices $u$ and $v$ from different clusters, let the probability of belonging to $\pi$ be $q$. We generate the graph $\boldsymbol{\pi}$ randomly, based on $p$ and $q$. We experiment with values of $k = 100$ and $n = 6$ for ease of visualization, embedding all points in a two-dimensional space. Each vertex's initial position originates from a normal distribution. In each iteration, we sample a subgraph of $\boldsymbol{\pi}$ uniformly, ensuring each vertex has an out-degree of $1$. We then optimize the corresponding vectors using InfoNCE loss with an SGD optimizer and iterate until convergence. Our experimental setup consists of an SGD learning rate of $1$, an InfoNCE loss temperature of $0.5$, and a batch size of $50$. We evaluate two scenarios with different $p$ and $q$ values: $p = 1$, $q = 0$, and $p = 0.75$, $q = 0.2$. The results of these experiments are visualized in Figure 3. The obtained embeddings exhibit the hallmark pattern of spectral clustering of graph $\boldsymbol{\pi}$.

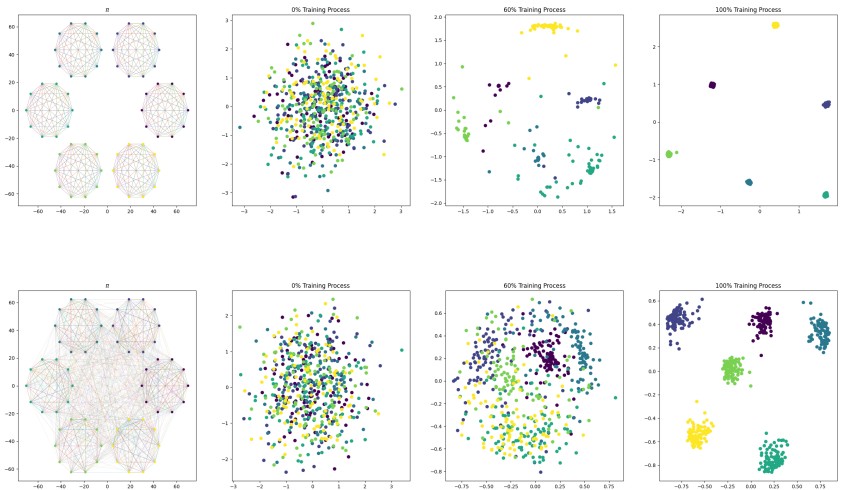

Figure 3: Visualizations of the optimization process using InfoNCE Loss on the vectors corresponding to $\pi$. Points of identical color belong to the same cluster within $\pi$. To showcase the internal structure of $\pi$, we randomly select 10 vertices from each cluster to display the edge distribution of $\pi$.

