# OpenReview forum: "Contrastive Learning is Spectral Clustering on Similarity Graph"
_ICLR.cc/2024/Conference — ICLR 2024 poster_

### Official Review · Reviewer_5UyX · 2023-10-20

**Soundness:** 3 good
**Presentation:** 3 good
**Contribution:** 3 good
**Rating:** 6
**Confidence:** 3

**Summary:**

This paper proves that contrastive learning with the standard InfoNCE loss is equivalent to spectral clustering on the similarity graph. Using this equivalence as the building block, the authors extend our analysis to the CLIP model and rigorously characterize how similar multi-modal objects are embedded together. Based on the maximum entropy principle, the authors demonstrate that the exponential kernels are the natural choices for capturing the local similarity structure for contrastive learning.

**Strengths:**

1. The  originality, quality, and significance are well supported by the proof of the equivalence of SimCLR and spectral clustering on the similarity graph and the extension to the multi-modal setting.

2. The clarity is satisfied based on the clear illustration of the analysis in Figure 1 and the clear motivations and contributions in the Introduction part of this paper.

**Weaknesses:**

1. The rationality of treating K_Z and \pi as MRFs and comparing the induced probability distributions on subgraphs should be better explaned, which is not well persuasive as shown in this version of the submitted paper.

2. For the definition of W, the authors are expected to further explain the mentioned unitary out-degree filter, which may be confused for the readers in understanding this definition.

3. The reason that cross-entropy loss can be converted to the combination of repulsion and attraction terms is expected to be further given after Lemma 2.4. Is it closely related to Lemma 2.5 and what is the specific relation?

4. In the experiment, the improvements of the proposed method is not obvious compared with SimCLR on the given datasets. The authors should further analyze the reason for 200 epochs and 400 epochs, respectively.

5. The authors should repeat each experiment for many times and list the mean and deviation to avoid the possible randomness, i.e.,  Table 1 and Table 4.

**Questions:**

1. Why choose Laplacian kernel and Simple Sum kernel for the MoCo experiment results should be further stressed, i.e., why the Gaussian kernel is not selected here.

2. Why the authors choose p=1,q=0 and p=0.75 and q=0.2 in the syntetic experiment?

---

> ### Author Response · Authors · 2023-11-18
> **Response to reviewer 5UyX**
>
> >Q1: The rationality of treating K_Z and \pi as MRFs and comparing the induced probability distributions on subgraphs should be better explaned, which is not well persuasive as shown in this version of the submitted paper.
>
> A1: Thank you for your suggestion. The reason why we compare $K_Z$ and $\pi$ by MRF are mainly two-folds. **These matrices are too large and, as they are constructed by the augmentation process, also make it hard to explicitly form them.** Using MRF and comparing the induced probability can serve as a good necessary condition to compare $K_Z$ and $\pi$, where the computation is not very costly by using the decomposition lemma 2.4.  We have added more explanations for this motivation in the revised manuscript.
>
> >Q2: For the definition of W, the authors are expected to further explain the mentioned unitary out-degree filter, which may be confused for the readers in understanding this definition.
>
> A2: Thank you for your suggestion. **We initially discussed the unitary out-degree filter right after the definition of $W$.** We have made changes in the revised manuscript to give clearer explanations to this defined term.
>
> >Q3: The reason that cross-entropy loss can be converted to the combination of repulsion and attraction terms is expected to be further given after Lemma 2.4. Is it closely related to Lemma 2.5 and what is the specific relation?
>
> A3: **The statement "cross-entropy loss can be converted to the combination of repulsion and attraction terms" is actually an intuitive summary of lemma 2.5.** We have rephrased the statement to make it directly point to lemma 2.5 in the revised manuscript. This statement is just naming the two terms in equation (4), the first term is an attraction and the second term a repulsion. These names are taken from the paper [1] as lemma 2.5 is just a slightly generalized result from this prior work.
>
> [1]: A Probabilistic Graph Coupling View of Dimension Reduction, Hugues Van Assel, Thibault Espinasse, Julien Chiquet, Franck Picard, NeurIPS 2022
>
> >Q4: In the experiment, the improvements of the proposed method is not obvious compared with SimCLR on the given datasets. The authors should further analyze the reason for 200 epochs and 400 epochs, respectively.
>
> A4: Thank you for your suggestion. **The experiments show that changing kernels may accelerate the convergence speed and improve classification accuracy.** Thus, we can see that changing kernel will usually result in an accuracy at 200 epochs that matches the accuracy of SimCLR at 400 epochs. When training proceeds, all methods exhibit an increase in accuracy. The reason why the proposed method is better than SimCLR may be intuitively understood by seeing our method as a sort of ensembles on kernel functions. The reason why the result of SimCLR is fairly good may be attributed to the good expressiveness of the Gaussian kernel.
>
> >Q5: The authors should repeat each experiment for many times and list the mean and deviation to avoid the possible randomness, i.e., Table 1 and Table 4.
>
> A5: Thank you for your suggestion. **We have repeated the experiments 3 times and updated the Tables in the revised manuscript.**
>
> >Q6: Why choose Laplacian kernel and Simple Sum kernel for the MoCo experiment results should be further stressed, i.e., why the Gaussian kernel is not selected here.
>
> A6: Thank you for your feedback. The reason why we choose Laplacian kernel and Simple Sum kernel is that it is consistent with what we did on the SimCLR experiments. **When the kernel is a Gaussian kernel, the loss is the initial MoCo loss, thus it is implicitly selected there.**
>
> >Q7: Why the authors choose p=1,q=0 and p=0.75 and q=0.2 in the syntetic experiment?
>
> A7: The goal of synthetic experiments is to give pictorial evidence of showing the equivalence of InfoNCE loss and spectral clustering. **The two sets of parameters  p=1,q=0 and p=0.75 and q=0.2 are chosen because they are typical.** p=1, q=0 depicts a scenario where points are fully connected and points among clusters are fully dis-connected. Thus a spectral clustering on this graph will have embeddings in the same clusters collapsed into a single point and different clusters have different collapse points. This phenomenon is clearly shown in the upper half of Figure 3. p=0.75 and q=0.2 shows another scenario where points have a much higher probability of another point in its cluster than connecting a point in another cluster. Thus, a spectral clustering on this graph will have embeddings that: the point from a same cluster will be closer and further from points in different clusters. This phenomenon is clearly shown in the lower half of Figure 3.

---

> ### Author Response · Authors · 2023-11-20
> **We would be grateful if you could take a look at the response**
>
> Dear Reviewer 5UyX:
>
> We sincerely appreciate your valuable time devoted to reviewing our manuscript. We would like to gently remind you of the approaching deadline for the discussion phase. We have diligently addressed the issues you raised in your feedback, providing detailed explanations. For instance, we add more explanations to some part of the paper according to your suggestions. We also conduct the experiments multiple times to record the standard deviation. Would you kindly take a moment to look at it?
>
> We are very enthusiastic about engaging in more in-depth discussions with you.

---

### Official Review · Reviewer_Hqsc · 2023-11-01

**Soundness:** 3 good
**Presentation:** 3 good
**Contribution:** 2 fair
**Rating:** 5
**Confidence:** 4

**Summary:**

- The paper proves that contrastive learning with the standard InfoNCE loss is equivalent to spectral clustering on the similarity graph, which is defined by the data augmentation process.
- The paper extends this result to the multi-modal setting and shows that CLIP is equivalent to spectral clustering on the pair graph.

**Strengths:**

- It provides a novel theoretical analysis of contrastive learning and its connection to spectral clustering, which can help understand the underlying mechanisms and principles of this popular self-supervised learning method.
- It proposes a new Kernel-InfoNCE loss with mixture of kernel functions that is inspired by theory and achieves better performance than the standard Gaussian kernel on several benchmark vision datasets

**Weaknesses:**

- I think the motivation is not good enough, such a conclusion is easy to obtain, i.e.,  InfoNCE loss is equivalent to spectral clustering. Since graph is pariwise relationship and constrastive is also pairwise relationship, both have the similar objective.
- as the first point, I think the kernel infoNCE is also not well motivated.

#### It's important to address these concerns regarding motivation in your paper. To improve the motivation for both the InfoNCE loss and the kernel InfoNCE, you might consider the following:

> InfoNCE Loss Motivation:

- Emphasize the practical significance and real-world applications of the InfoNCE loss. How does it relate to real-world problems or datasets in a way that goes beyond spectral clustering?
- Highlight specific challenges or limitations in existing methods that the InfoNCE loss aims to address.

> Kernel InfoNCE Motivation:

- Explain how the kernel InfoNCE extends the motivation from the InfoNCE loss. What specific problems or scenarios does the kernel-InfoNCE address that are not covered by the standard InfoNCE?
- Provide examples or use cases where kernel InfoNCE can be especially valuable.
> By offering a more compelling rationale and demonstrating the practical relevance of these concepts, you can strengthen the motivation for these components in your paper.

**Questions:**

> see the Weaknesses
- Could you please share your reasons behind this?  it to be innovative?

---

> ### Author Response · Authors · 2023-11-18
> **Response to reviewer Hqsc**
>
> >Q1: I think the motivation is not good enough, such a conclusion is easy to obtain, i.e., InfoNCE loss is equivalent to spectral clustering. Since graph is pariwise relationship and constrastive is also pairwise relationship, both have the similar objective.
>
> A1: We are glad that you agree that our conclusion is evidently correct, but providing rigorous proof for this matter is not a trivial task. In fact, there hasn't been any proof showing the exact equivalence of SimCLR (InfoNCE loss) and spectral clustering. There have been analyses on variants of InfoNCE loss (spectral contrastive loss [1], euclidean MSE variant of SimCLR[2]), notably [1] showing a variant of InfoNCE loss is spectral clustering and receives oral in NIPS 2021. **However, as InfoNCE loss is the most popular contrastive loss used in literature, we fill the theoretical gap that InfoNCE is also conducting spectral clustering. The motivation for our work is to mainly clarify the equivalence of the initial InfoNCE loss (not its variant) and spectral clustering, which is meaningful and the techniques we used are also not trivial.**
> Therefore, it is important to emphasize our main contribution, which is the rigorous proof of the exact equivalence between contrastive learning (SimCLR) and spectral clustering. While this may be easily observed in the synthetic experiments we conducted, experimental observations cannot fully replace rigorous mathematical proofs. For instance, the simplex method is a widely used algorithm that was proven to be polynomial in the probabilistic sense later than the invention of the algorithm.
>
> [1] Provable Guarantees for Self-Supervised Deep Learning with Spectral Contrastive Loss, Jeff Z. HaoChen, Colin Wei, Adrien Gaidon, Tengyu Ma, NeurIPS 2021 (Oral)
>
> [2] Contrastive and Non-Contrastive Self-Supervised Learning Recover Global and Local Spectral Embedding Methods, Randall Balestriero, Yann LeCun, NeurIPS 2022
>
> >Q2: as the first point, I think the kernel infoNCE is also not well motivated.
>
> A2: The motivations for introducing kernel infoNCE are two-fold. **One is that this loss can be easily derived from our MRF framework by changing kernels.** The other is that the choices of exponential kernels (which are used in the loss) can be understood by a maximal entropy principle.
>
> >Q3: It's important to address these concerns regarding motivation in your paper. To improve the motivation for both the InfoNCE loss and the kernel InfoNCE, you might consider the following......, InfoNCE Loss Motivation:......, Kernel InfoNCE Motivation:......
>
> A3: Thank you for your suggestion. We did not propose InfoNCE loss ourselves, InfoNCE loss is itself one of the most popular losses used in contrastive learning. **The motivation of our paper is to rigorously prove its equivalence to spectral clustering, which fills the gap in the area of theoretical understanding of self-supervised learning algorithms.** The motivation for our kernel-InfoNCE loss is that we show that it can be seen as a natural generalization of InfoNCE loss from our MRF and maximal entropy theory in sections 3 and 5. Moreover, we test the effectiveness of kernel-InfoNCE loss empirically.

---

> ### Author Response · Authors · 2023-11-20
> **We would be grateful if you could take a look at the response**
>
> Dear Reviewer Hqsc:
>
> We sincerely appreciate your valuable time devoted to reviewing our manuscript. We would like to gently remind you of the approaching deadline for the discussion phase. We have diligently addressed the issues you raised in your feedback, providing detailed explanations. For instance, we make more explanations of the motivation of our paper. We also discuss the motivation of the proposed kernel-InfoNCE loss. Would you kindly take a moment to look at it?
>
> We are very enthusiastic about engaging in more in-depth discussions with you.

---

### Official Review · Reviewer_TB2Y · 2023-11-01

**Soundness:** 3 good
**Presentation:** 2 fair
**Contribution:** 3 good
**Rating:** 6
**Confidence:** 4

**Summary:**

The paper applies a probabilistic graph coupling perspective to view two typical constrastive learning methods, including SimCLR and CLIP, and interpretes them as spectral clustering or generalized spectral clustering. Moreover, it also attempts to propose to use exponential kernels to replace the Gaussian kernel. Preliminary experiments show that using a mixtures of exponential kernels to replace the Gaussian kernel in the SimCLR loss yields improved classification accuracy.

**Strengths:**

+ It is interesting to interprete the InfoNCE loss in SimCLR and CLIP into the perspective of probablistic graph coupling and thus find the connection to spectral clustering or generalized spectral clustering.

**Weaknesses:**

- The reviewer was confused by the discussion before introducing problem (P1). Since that it is required that the $\mathbf \alpha$ has fewer nonzero entries, some objective of sparsity-promoting property is necessary. However, in (P1) an entropy regularization term is imposed. It is well known that the optimal solution for the maximal entropy problem in the discrete random variable is a uniform distribution. Here, the optimal solution for $\alpha_i$ should be $1/n$. It is weired to have a problem in (P2) and the solution in Theorem 5.1. Note that $\tau$ is the Lagrangian multiplier, i.e., dual variable, it is incomplete to have the dual variable inside.

- Moreover, there are mistakes in the formulation of (P2). It is neither the Lagrangian nor the Lagrangian dual problem. It is misleading to claim minimizing (P2) producing an upper bound of (P1).

- In Section 5.2, it is stated that Theorem 5.1 suggests that the loss function of that form is a natural choice for characterizing the neighborhood similarity structure. The reviewer cannot see this point. Such a form is nothing but a choice on purpose to use the maximal entropy (or due ot mistakes?).

- In Eq. (6), it is a RBF kernel, cannot be directly yielded from the form in Theorem 5.1. Because having an exponential form does not imply to have the property of a RBF kernel. In this way, the so-called kernel-InforNCE is nothing but a heuristic form to define the similarity in the InfoNCE loss function.

- The related work is not good. Some remarks on the previous work are either improper or even misleading.

- The experimenal evaluation is limited.

**Questions:**

- The reviewer was confused by the discussion before introducing problem (P1). It is weired to have a problem in (P2) and the solution in Theorem 5.1.

- Moreover, there are mistakes in the formulation of (P2). It is neither the Lagrangian nor the Lagrangian dual problem. It is misleading or something is missig to claim minimizing (P2) producing an upper bound of (P1).

- The reviewer cannot see that ``Theorem 5.1 suggests that the loss function of that form is a natural choice for characterizing the neighborhood similarity structure".

- The reviewer is not clear how to have a RBF kernel from the form in Theorem 5.1.

---

> ### Author Response · Authors · 2023-11-18
> **Total response to reviewer TB2Y**
>
> Thank you very much for your detailed review of our paper. We have noticed that you mainly have some confusion regarding our analysis in Section 5, and we would be able to clarify these confusions effectively.
>
> First of all, let's discuss the idea behind our optimization problem (P1). In fact, $\psi_1$ represents the similarity between the query sample q and its positive sample. **In contrastive learning, we certainly want this similarity to be maximal among contrastive samples**. Therefore, we have the following 3 constraints: $\alpha^T 1_n = 1$ and $\psi_1 - \sum \alpha_i \psi_i \leq 0$ and $\alpha \geq 0$. It is important to note that the second constraint alone does not directly imply that $\psi_1$ is maximized. Let's consider a few examples, where there will be 3 contrastive samples and the first one is a positive sample. We abbreviate $\psi = (\psi_1, \psi_2, \psi_3)$:
>
> Example 1:
> $\psi = (0.8, 0.5, 0.5)$
>
> In this case, there is a unique solution for $\alpha$, which is (1, 0, 0). In fact, if $\psi_1$ is maximal among contrastive samples $\alpha$ must have a unique solution and should not be a uniform distribution. In this scenario, H($\alpha$)=0.
>
> Example 2:
> $\psi = (0.8, 0.8, 0.5)$
>
> In this case, there are multiple solutions for $\alpha$, such as (1, 0, 0), (0, 1, 0), (0.3, 0.7, 0), (0.5, 0.5, 0). The optimal solution for (P1) is (0.5, 0.5, 0), and in this case, H($\alpha$)= $\log 2$. Note that $\alpha$ is still not a uniform distribution. Instead, it is a uniform distribution among the first two elements.
>
> Example 3:
> $\psi = (0.8, 1, 0)$
>
> In this case, there are multiple solutions for $\alpha$ as well, such as (0, 0.8, 0.2), (0.2, 0.64, 0.16), and so on. The optimal solution for (P1) is not straightforward to compute.
>
> **Therefore, in general, the optimal solution to (P1) is not obtained when $\alpha$ is a uniform distribution**. Moreover, as observed from the examples above, as long as $\psi_1$ is not the maximal among contrastive samples, H($\alpha$) will always be greater than 0. Hence, we leverage this characteristic and attempt to optimize $\psi$ in a way that H($\alpha$) tends to approach 0 (because H((1, 0, 0, 0, ...))=0).
>
> Therefore, if we can solve the program $min_\psi \max_\alpha H(\alpha)$  (where the max should be in fact optimization problem (P1), but we abbreviate the constraints  $\psi_1 - \sum \alpha_i \psi_i \leq 0$ and $\alpha^T 1_n = 1$ and $\alpha \geq 0$ for notation simplicity here), we can achieve our objective in contrastive learning. **However, contrastive learning does not directly use this program because $\max_\alpha H(\alpha)$, which is (P1), is not easy to compute**.
>
> So, we consider transforming (P1) as follows:
> We introduce a function $L(\alpha, \tau) = H(\alpha) - 1/\tau (\psi_1 - \sum \alpha_i \psi_i)$ and impose the constraints $\alpha^T 1_n = 1$ and $\alpha \geq 0$.
> We can observe that for any $\tau > 0$, $L(\alpha, \tau) \geq$ the objective function of (P1). This is because for any $\alpha$ that satisfies (P1), we have $\psi_1 - \sum \alpha_i \psi_i \leq 0$. Therefore, $H(\alpha) - 1/\tau (\psi_1 - \sum \alpha_i \psi_i) \geq H(\alpha)$.
>
> Hence, we can consider the following program: $max_\alpha L(\alpha, \tau)$ subject to $\alpha^T 1_n = 1$ and $\alpha \geq 0$. We know that the solution to this problem is an upper bound of the original problem.
>
> **The objective of (P2) is $-E(\alpha) = \tau L(\alpha, \tau)$.** Thus, from $\tau (H(\alpha) - (\psi_1 - \sum \alpha_i \psi_i)) \geq \tau (H(\alpha))$. Thus, (P2) is an $\tau$ times upper bound of (P1). The way we derive the above bounding property is known as weak duality in Lagrange duality theory (https://en.wikipedia.org/wiki/Duality_(optimization)).
>
> Notably, our Theorem 5.1 proves that (P2) has a closed-form solution, which is $-\tau \log \frac{\exp (\frac{1}{\tau} \psi_{1})}{\sum_{i=1}^n \exp (\frac{1}{\tau} \psi_{i})}$. Therefore, we know that $-\log \frac{\exp (\frac{1}{\tau} \psi_{1})}{\sum_{i=1}^n \exp (\frac{1}{\tau} \psi_{i})}$ is an upper bound of (P1). **From this perspective, a loss form like InfoNCE is natural for solving contrastive learning problems.**

---

> > ### Comment · Reviewer_TB2Y · 2023-11-22
> >
> > The reviewer appreciates the great efforts in providing the clarification. After reading the rebuttal, and the revised manuscript, the raised major concerns or technical issues have been resolved.  Anyway, for the revised manuscript, the reviewer would like to also give a revised rating, though the related work and experiments are still not good enough.

---

> ### Author Response · Authors · 2023-11-18
> **Response 1 to reviewer TB2Y**
>
> >Q1: The reviewer was confused by the discussion before introducing the problem (P1). Since that it is required that the \alpha has fewer nonzero entries, some objective of sparsity-promoting property is necessary. However, in (P1) an entropy regularization term is imposed. It is well known that the optimal solution for the maximal entropy problem in the discrete random variable is a uniform distribution. Here, the optimal solution for \alpha_i should be 1/n. It is weird to have a problem in (P2) and the solution in Theorem 5.1. Note that \tau is the Lagrangian multiplier, i.e., dual variable, it is incomplete to have the dual variable inside.
>
> A1: **As we have explained above through examples, the optimal solution to the maximal entropy problem is 1/n when the constraint of $\psi_1 - \sum \alpha_i \psi_i \leq 0$ does not exist.** However, this may not be the case where this constraint is added. The Lagrangian function $E(\alpha, \tau)$ is generated by introducing dual variable $\tau$. In the initial manuscript, we abbreviated the dependency of E on $\tau$ for notation simplicity. The problem (P2) is $max_{\alpha} E(\alpha, \tau)$, which calculates the objective function used in a Lagrangian dual problem of (P1) where $\tau$ is the dual variable.
>
> >Q2: Moreover, there are mistakes in the formulation of (P2). It is neither the Lagrangian nor the Lagrangian dual problem. It is misleading to claim minimizing (P2) producing an upper bound of (P1).
>
> A2: The formulation of (P2) is calculating the **objective function** in the Lagrangian dual problem of (P1). **As we discussed in the above total response, (P2) provides a $\tau$ times upper bound for the initial problem (P1)**. Then minimizing (P2) will make (P1) smaller.
>
> >Q3: In Section 5.2, it is stated that Theorem 5.1 suggests that the loss function of that form is a natural choice for characterizing the neighborhood similarity structure. The reviewer cannot see this point. Such a form is nothing but a choice on purpose to use the maximal entropy (or due ot mistakes?).
>
> A3: **As we have discussed in the above total response, the value of H($\alpha$) reflects how many contrastive samples have similar similarity to the query sample compared to the similarity of the positive sample and query sample**. Thus the optimal value of the optimization problem (P1) can characterize the neighborhood similarity structure. Then we show that the loss function of the form in Theorem 5.1 can bound the problem (P1), in this way we explain why the loss function of the form of InfoNCE is natural.
>
> >Q4: In Eq. (6), it is an RBF kernel, that cannot be directly yielded from the form in Theorem 5.1. Because having an exponential form does not imply having the property of an RBF kernel. In this way, the so-called kernel-InforNCE is nothing but a heuristic form to define the similarity in the InfoNCE loss function.
>
> A4: That is a great question! **There are various different ways to define the similarities between two points, and in (6), we pick the similarity function $\psi_i = C - \| f(q) - f(p_i) \|^{\gamma}$, where C is a large positive constant.** The form in theorem 5.1 is exactly the exponential kernels given by equation (6). Thus our derivation is not a heuristic.
>
> >Q5: The related work is not good. Some remarks on the previous work are either improper or even misleading.
>
> A5: Thank you for your suggestions. **We have added more summaries of existing works and made the contributions of previous works more accurate.**
>
> >Q6: The experimenal evaluation is limited.
>
> A6: We have conducted experiments by changing kernels based on SimCLR and MoCo. **The experiments show improvements upon baselines, which support our theory that the kernel function may be changed.**  As we construct kernels using the exponential family kernel, further validate our derivation that the exponential kernel is natural. We further conduct synthetic experiments to give a pictorial understanding of our main theorem: contrastive learning is spectral clustering.

---

> ### Author Response · Authors · 2023-11-18
> **Response 2 to reviewer TB2Y**
>
> >Q7: The reviewer was confused by the discussion before introducing problem (P1). It is weired to have a problem in (P2) and the solution in Theorem 5.1.
>
> A7: **As we pointed out earlier in the total response, direct solving (P1) is hard. Therefore, we propose a transformation of (P1) as follows**: We introduce the function $L(\alpha, \tau) = H(\alpha) - \frac{1}{\tau} (\psi_1 - \sum \alpha_i \psi_i)$ and impose the constraints $\alpha^T 1_n = 1$ and $\alpha \geq 0$.
> We observe that for any $\tau > 0$, $L(\alpha, \tau) \geq$ the objective function of (P1). This is because any \alpha that satisfies (P1) also satisfies $\psi_1 - \sum \alpha_i \psi_i \leq 0$. Therefore, $H(\alpha) - \frac{1}{\tau} (\psi_1 - \sum \alpha_i \psi_i) \geq H(\alpha)$.
> Hence, we can consider the following program: $\max_\alpha L(\alpha, \tau)$ subject to $\alpha^T 1_n = 1$  and $\alpha \geq 0$. The solution to this problem provides an upper bound for the original problem.
> Consequently, we conclude that the objective of (P2), and (P2) can be analytically solved and give the solution in Theorem 5.1.
>
> >Q8: Moreover, there are mistakes in the formulation of (P2). It is neither the Lagrangian nor the Lagrangian dual problem. It is misleading or something is missig to claim minimizing (P2) producing an upper bound of (P1).
>
> A8: The introduction of the dual variable $\tau$ leads to the generation of the Lagrangian function $E(\alpha, \tau)$. **In the original manuscript, we simplified the notation by omitting the explicit dependency of E on $\tau$.** The problem (P2) corresponds to maximizing $E(\alpha, \tau)$, which represents the objective function in the Lagrangian dual problem of (P1), where \tau serves as the dual variable.
>
> >Q9: The reviewer cannot see that "Theorem 5.1 suggests that the loss function of that form is a natural choice for characterizing the neighborhood similarity structure".
>
> A9: **As mentioned earlier in the total response, the value of H($\alpha$) provides insight into the similarity between the query sample and contrastive samples, relative to the similarity between the query sample and the positive sample.** Consequently, the optimal solution of the optimization problem (P1) captures the neighborhood similarity structure. By demonstrating that the loss function in Theorem 5.1 can provide an upper bound for the problem (P1), we can explain the natural suitability of the InfoNCE loss function.
>
> >Q10: The reviewer is not clear how to have a RBF kernel from the form in Theorem 5.1.
>
> A10: Multiple approaches exist for defining the similarities between two points, and in equation (6), we specifically choose the similarity function $\psi_i = C - \| f(q) - f(p_i) \|^{\gamma}$, where C represents a large positive constant. The form presented in theorem 5.1 corresponds precisely to the exponential kernels described by equation (6). **When $\gamma=2$, this recovers the RBF kernel, which is a special case of exponential kernel.**

---

> ### Author Response · Authors · 2023-11-20
> **We would be grateful if you could take a look at the response**
>
> Dear Reviewer TB2Y:
>
> We sincerely appreciate your valuable time devoted to reviewing our manuscript. We would like to gently remind you of the approaching deadline for the discussion phase. We have diligently addressed the issues you raised in your feedback, providing detailed explanations. For instance, we make more detailed explanations about how to derive optimization problem (P1) through examples and derivations. We also give more discussions on the bounding relationship between (P1) and (P2). Would you kindly take a moment to look at it?
>
> We are very enthusiastic about engaging in more in-depth discussions with you.

---

### Official Review · Reviewer_PnKu · 2023-11-01

**Soundness:** 4 excellent
**Presentation:** 3 good
**Contribution:** 4 excellent
**Rating:** 10
**Confidence:** 3

**Summary:**

The authors present a theoretical result concerning contrastive learning. Contrastive learning is a semi-supervised task that aims to map objects into an embedding space such that similar objects are close and dissimilar objects are far apart. Their results concern the widely-used SimCLR loss, an example of an InfoNCE loss. The authors show that optimizing InfoNCE is equivalent to solving a spectral clustering problem. Based on this theoretical insight, they give an argument that exponential kernels are natural and propose a variant of InfoNCE, Kernel-InfoNCE, where they use an alternative exponential kernel in place of the usual Gaussian kernel. Doing so led them to using a Simple Sum kernel, which achieves slightly improved empirical performance on CIFAR image-text data sets.

This paper is closely related to HaoChen et al 2021; that paper proposed a type of contrastive loss that constitutes performing spectral clustering. The authors of this paper extend this by proving that SimCLR itself constitutes performing spectral clustering. This paper is also related to Van Assel et al., 2022, which analyzes dimensionality reduction methods such as t-SNE using a Markov random field framework adopted by the authors of this paper.

**Strengths:**

This is an important theoretical result concerning a widely-used method. This work helps bridge the gap between theory and practice in contrastive learning. I have several comments, none of which are major.

**Weaknesses:**

Minor comments:

I found parts of the text difficult to follow because it lacks guideposts explaining the purpose of each section at a high level.

It would help to have a definition of spectral clustering for the purposes of the paper.

Eq1: I think this is meant to be a sum over different q's; the text says as much, but that's not how it's defined in Eq1.

"we will flip $X_i$ to get $X_j$ with probability 1/9; it's not clear what "flip" means or why the probability is 1/9.

**Questions:**

See above

---

> ### Author Response · Authors · 2023-11-18
> **Response to reviewer PnKu**
>
> > Q1: I found parts of the text difficult to follow because it lacks guideposts explaining the purpose of each section at a high level.
>
> A1: Thank you for your suggestion. We have added a brief summary of the purposes of each section at the beginning of each section in the revised manuscript.
>
> >Q2: It would help to have a definition of spectral clustering for the purposes of the paper.
>
> A2: Thank you for your comment. We have added the definition of spectral clustering as definition 2.7 in the revised version of our manuscript.
>
> >Q3: Eq1: I think this is meant to be a sum over different q's; the text says as much, but that's not how it's defined in Eq1.
>
> A3: We are sorry for any confusion here, what we mean is that the total loss is a summation of query samples (i.e. over q's).
>
> >Q4: "we will flip $X_i$ to get $X_j$ with probability 1/9; it's not clear what "flip" means or why the probability is 1/9.
>
> A4: This sentence is just an example to intuitively understand what $\pi$ consists of. It does not mean the actual number is 1/9. The actual meaning of this sentence is that "suppose $X_j$ has a probability, say 1/9, to be an augmentation of an object $X_i$".

---

### Author Response · Authors · 2023-11-18
**General Response**

We would like to express our sincere gratitude to the reviewers and the area chair for their diligent efforts in reviewing our paper. We greatly appreciate their insightful comments and suggestions.
Based on their feedback, we have made updates to the paper:

1. Adding more descriptions about the motivation of each section.
2. Adding explanations to some derivations in the paper to make them more understandable.
3. Adding standard deviation to the experiment results.
4. Correct some typos.

---

### Meta-Review · Area_Chair_fRiV · 2023-12-11

**Metareview:**

Thanks for your submission to ICLR.

This paper takes a mostly theoretical look at contrastive learning, and demonstrates an equivalence between constrastive learning with InfoNCE loss and a spectral clustering-type optimization problem.  They also show how this extends to CLIP, and further how this result suggests other extensions of the InfoNCE loss.

Reviewers were generally positive about the paper, though some of the reviewers noted some issues with the readability, motivation, and related work section of the paper.  The authors responded to these, and ultimately 3 of the 4 reviewers were left positive (one very positive) at the end of the discussion period.

I also took a look at this paper.  While I did not completely check every proof, I did read through this quite carefully and did not see any issues with the overall mathematical derivations.  I also found the connections to be interesting.  My only issue with the paper---and I think this is fixable but important---is that I take some issue with the use of the term "spectral clustering" in this paper.  The given definition of spectral clustering, equation 5, is not the usual definition of spectral clustering.  (I used to work extensively in that area many years ago.)  Typically, spectral clustering is defined as a relaxation of a clustering problem (e.g., a graph cut of some kind), where one seeks a partition matrix Z.  Because that problem is usually NP-hard to solve, one relaxes Z to be an orthogonal matrix, and then the solution can be obtained by eigenvectors of the Laplacian (or related matrix).  In the current paper, the optimization problem as given is not actually solving such a relaxation, as the Z matrix is unconstrained here, and there is an addition of a regularizer instead.  So what the authors are claiming is spectral clustering is actually something else, something that I don't think involves a spectral decomposition at all.

I still think the result is interesting, and worth publishing, but I don't like the terminology here that this is equivalent to spectral clustering.  (Or, perhaps I am missing something, but in that case, the authors should be sure to clarify the paper on this point.)  I would ask the authors to clarify this point in a final draft.

**Justification For Why Not Higher Score:**

The paper has an interesting result, but it's not the most clearly-written paper.  Several reviewers noted this, and I also noted issues with the authors' writing.

**Justification For Why Not Lower Score:**

Three of the four reviewers are happy to recommend acceptance for the paper, with one even giving it a 10/10.

---

### Decision · Program_Chairs · 2024-01-16

Accept (poster)